# Potential Role of Macrophage Polarization in the Progression of Hunner-Type Interstitial Cystitis

**DOI:** 10.3390/ijms25020778

**Published:** 2024-01-08

**Authors:** Kwang Jin Ko, Gahyun Kim, Hyun Hwan Sung, Woong-Yang Park, Kyu-Sung Lee

**Affiliations:** 1Department of Urology, Samsung Medical Center, Sungkyunkwan University School of Medicine, Seoul 06351, Republic of Korea; hyunhwan.sung@samsung.com; 2Samsung Genome Institute, Samsung Medical Center, Seoul 06351, Republic of Korea; ghmdsr@gmail.com (G.K.); woongyang.park@samsung.com (W.-Y.P.); 3Samsung Advanced Institute of Health Science and Technology, Sungkyunkwan University, Seoul 06351, Republic of Korea; 4Research Institute for Future Medicine Samsung Medical Center, Seoul 06351, Republic of Korea

**Keywords:** autoimmune, interstitial cystitis, macrophage, RNA-seq, transcriptome analysis

## Abstract

Background: Hunner-type interstitial cystitis (HIC) is a chronic inflammatory condition of the bladder. However, it remains unclear whether there is a causal relationship between the presence of Hunner lesions and seemingly normal-appearing areas in the bladder (non-Hunner lesions). This study aimed to investigate the fundamental aspects of HIC by examining potential genetic differences between Hunner and non-Hunner lesions and elucidate their role as potential markers in the progression and suppression of the disease. Methods: This cross-sectional study enrolled patients with HIC (*n* = 10) who underwent supratrigonal cystectomy along with augmentation cystoplasty. Full-thickness bladder tissue was collected from Hunner and non-Hunner lesions in the same patient. Normal bladder tissue biopsies were also obtained as controls. Whole transcriptome analysis was performed to analyze the gene expression patterns and immune cell populations. Results: The mucosal layers of patients exhibited similar pathway dysregulation across Hunner and non-Hunner lesions, with immunerelated pathways being prominently affected. In the mucosal layer, genes related to anti-inflammatory and immune suppression were downregulated in Hunner lesions compared to non-Hunner lesions. Moreover, in Hunner lesions, genes related to macrophage differentiation and polarization, such as *VSIG4*, *CD68*, *MAFB*, and *LIRB4*, were downregulated. The cell fraction of M2 macrophages was found to decrease in Hunner lesions. Immunohistochemical staining revealed an elevated fraction of M1 macrophages and a reduced fraction of M2 macrophages in Hunner lesions compared to those in non-Hunner lesions. In the muscular layer, transcriptomic evidence of muscle thickness was observed in both Hunner and non-Hunner lesions; however, the difference was not significant. Conclusion: Hunner lesions showed a reduced expression of anti-inflammatory and immunosuppressive factors compared to non-Hunner lesions, along with alterations in immune cell populations. This study suggests the possibility that macrophage polarization is related to the progression from non-Hunner lesions to Hunner lesions, suggesting its relevance to the characteristics of autoimmune diseases.

## 1. Introduction

Interstitial cystitis (IC) is a chronic condition characterized by pelvic pain and pressure or discomfort related to bladder filling. Unfortunately, the etiology of IC is not well understood, and no curative treatments are currently available. IC has recently been classified into IC with Hunner lesions (Hunner-type IC, HIC) and without Hunner lesions (bladder pain syndrome) [1,2]. In HIC, the histopathological characteristics of Hunner lesions include chronic inflammation of the bladder mucosa, epithelial denudation, and subepithelial mastocytosis [3]. Notably, these histological changes can be observed in the entire bladder and are not confined to Hunner lesions [4].

Transcriptomic profiling, which involves the analysis of gene expression patterns, can provide insights into the molecular mechanisms underlying diseases, including HIC, and may lead to the identification of novel therapeutic targets. Several previous studies have used transcriptomic profiling to investigate the pathogenesis of HIC, revealing alterations in various signaling pathways, including inflammation, immune response, and neuroplasticity [5,6,7,8,9]. However, most of these studies have analyzed only bladder mucosal tissues from cold-cup biopsies or public databases and have inconsistent limitations. Owing to the complex structure of the bladder, the distribution of nerve and immune cells in each layer is diverse, and the mechanism of cellular interactions within the layers is ideal for explaining the underlying pathogenesis. However, to date, no study has used full-thickness bladder tissues obtained after partial cystectomy in patients with HIC.

The present study aimed to investigate the fundamental aspects of HIC by examining potential genetic differences between Hunner and non-Hunner lesions and reveal their role as potential markers in the progression and suppression of the disease. We performed transcriptomic profiling of the full layer of bladder tissue of Hunner lesions and non-Hunner lesions in patients with HIC and compared them with bladder tissue from controls.

## 2. Results

### 2.1. Patient Characteristics

Table 1 shows the baseline characteristics of the patients in this study. The patients underwent an average of two (range 0–5) endoscopic ablations before partial cystectomy, and the median maximal cystometric capacity was 160.0 cc.

### 2.2. Fibrosis and Immune Activation in the Mucosal Layer of Patients with Hunner Type Interstitial Cystitis

Whole transcriptome sequencing (WTS) was performed on mucosal tissue samples from both Hunner lesions and non-Hunner lesions of ten patients with HIC and four healthy donors.

To explore the transcriptomic characteristics of the mucosal layer in patients with IC, we first performed GSVA. Several pathways associated with the immune system, including interferon-gamma response (*p* = 0.002), inflammatory response (*p* = 0.005), allograft rejection (*p* = 0.006), interferon-alpha response (*p* = 0.010), complement system (*p* = 0.004), TNF-α signaling via NF-kB (*p* = 0.032), IL2/STAT5 signaling (*p* = 0.008), and IL6/JAK/STAT3 signaling (*p* = 0.003) exhibited significant variations in GSVA scores among Hunner lesion, non-Hunner lesion, and normal groups. These pathways exhibited similar GSVA scores between Hunner lesions and non-Hunner lesions (Figure 1B,C). Epithelial-mesenchymal transition (EMT) showed a tendency to be upregulated in Hunner/non-Hunner lesions compared to normal tissue, although it was not statistically significant. In contrast, pathways related to peroxisomes, fatty acid metabolism, and adipogenesis were downregulated in the mucosa of patients with HIC (Figure 1B,C). For most genes, there was a high correlation between their expression in Hunner and non-Hunner lesions compared to that in healthy tissues (r = 0.934, *p* < 0.001). We observed the upregulation of immunoglobulin genes, including CXCL6 chemokine, CHI3L1, POU4F1, MMP10, and IL11 (Figure 1D, Appendix A). Also, as expected, the expression of epithelial marker genes was decreased in Hunner and non-Hunner lesions (Appendix A). In agreement with this transcriptomic difference, we confirmed fibrosis in non-Hunner and Hunner lesions using hematoxylin and eosin staining (Figure 1E).

### 2.3. Reduced Anti-Inflammation and Immunosuppression in the Mucosal Layer of Hunner Lesions Compared to Non-Hunner Lesions

After conducting a DEG analysis between Hunner and non-Hunner lesions in the mucosal layer, we performed GSEA. Pathways related to fibrogenesis, such as positive regulation of fibroblast proliferation, collagen catabolic processes, and TGF-β signaling, were enriched in Hunner lesions. In contrast, we found that the most enriched pathways in non-Hunner lesions were related to the negative regulation of the immune system (Figure 2A).

A dysregulated immune system is a key feature of chronic diseases, and various cell types participate in the dysregulated system. We focused on the negative regulation of the immune system. Genes associated with negative regulation of the immune system were mapped using a volcano plot, which revealed that most genes were downregulated in Hunner lesions (Figure 2B). Among them, some significantly downregulated genes, such as *VSIG4* and *MAFB*, are mainly expressed in macrophages or are involved in macrophage polarization (Figure 2B). We performed a bulk RNA-seq deconvolution analysis [10] to estimate the immune cell populations and found that the cell fraction of M2 macrophages was decreased in Hunner lesions, whereas that of M1 macrophages increased (Figure 2C). Immunohistochemical staining showed that M1 and M2 macrophages were both increased in Hunner lesions, which was partially consistent with the transcriptomic analysis (Figure 2D, left). Notably, an increased ratio of M1 macrophages to M2 macrophages (“M1/M2 ratio”) was observed in Hunner lesions (Figure 2D, right). Cell counts (numbers per HPF) of M1 (CD80 and MHC II) and M2 (CD163 and CD206) macrophages in Hunner and non-Hunner lesions are shown in Appendix A.

We also compared genes encoding cytokines and growth factors. Two interleukin genes, *IL19* and *IL11*, were significantly upregulated in Hunner lesions. However, some genes were significantly downregulated in Hunner lesions compared to non-Hunner lesions. These genes included chemokines (*CXCL16*, *CXCL9*, and *CCL25*), interleukins (*IL13* and *IL12B*), tumor necrosis factors (*LTA* and *TNFSF13*), and *IL1RN*, which encodes the interleukin 1 receptor antagonist protein (Figure 2E).

### 2.4. Thickening of Serosal/Muscular Layer of Patients with Interstitial Cystitis

To decipher the pathological mechanisms of HIC in the serosa and muscle layers, we first performed a DEG analysis between Hunner/non-Hunner lesions and normal tissues. Similar to our observation in the mucosal layer, the gene expression patterns of Hunner/non-Hunner lesions compared to those in healthy tissues were highly correlated (r = 0.953, *p* < 0.001) (Figure 3A); 75 genes were significantly upregulated in both Hunner and non-Hunner lesions. These upregulated genes included genes related to fibroblast growth factor (FGF7, FGF2, and FGFR1OP2), indicating the progression of fibrosis into the serosa/muscle layer beyond the mucosal layer (Figure 3B). We also identified some dysregulated pathways, including those involved in fibrosis and the immune response, in patient groups compared to those in the healthy group. Notably, pathways related to muscle development were enriched, implying that reduced bladder capacity, one of the symptoms of the disease, may be caused by smooth muscle hyperplasia/hypertrophy (Figure 3C). Additionally, we observed enrichment of genes related to muscle hypertrophy and the smooth muscle cell proliferation pathway in Hunner lesions compared to healthy tissues (Figure 3D); however, this result was not significant.

## 3. Discussion

In the present study, in patients with HIC, both Hunner and non-Hunner lesions were associated with dysregulated pathways related to inflammatory and immune responses compared to those in healthy bladder tissue, which is consistent with the results of previous studies. In addition, several IC animal model studies have reported that inflammatory responses and bladder fibrosis are alleviated through the overexpression or downregulation of specific miRNAs involved in the JAK/STAT signaling pathway [11,12]. Consistent with these studies, our transcriptomic analysis identified the IL6/JAK/STAT3 signaling pathway, which is aberrantly hyperactivated in patients with chronic inflammatory and autoimmune diseases, as one of the enriched pathways of innate immunity. We also observed upregulation of immunoglobulin genes, including the chemokine *CXCL6*; *CHI3L*, which is strongly associated with diseases such as liver fibrosis, rheumatoid arthritis, and inflammatory bowel disease [13,14,15]; *POU4F1*, which is a neural transcription factor; *MMP10*, which contributes to extracellular matrix remodeling; and *IL11*, which has been suggested as a therapeutic target in fibrotic diseases [16,17]. In contrast, the peroxisome, fatty acid metabolism, and adipogenesis pathways were downregulated in patients with HIC. Peroxisomes are key regulators of immune functions and inflammation during development and infection. Peroxisome deficiency is also concomitant with an increase in free fatty acids, which are drivers of inflammation [18]. Peroxisome deficiency induces liver fibrosis and idiopathic pulmonary fibrosis [19,20]. Excessive *CHI3L1 (YKL*-*40)* production may cause matrix accumulation, leading to tissue fibrosis [21]. Moreover, serum and urine YKL-40 levels are higher in patients with IC and detrusor fibrosis than in those without it [22]. Collectively, these previous findings and ours suggest that detrusor fibrosis is a result of CHI3LI upregulation and peroxisome pathway downregulation in patients with HIC.

Many previous studies have focused only on the characteristics of Hunner lesions compared to healthy tissues [5,6,7,23,24]. In the present study, we broadened our understanding of IC progression by comparing Hunner lesions with non-Hunner lesions in the same patients. We believe that non-Hunner lesions ultimately become Hunner lesions. Notably, we observed differences in the immune responses of Hunner and non-Hunner lesions, which can prevent the transformation of non-Hunner lesions to Hunner lesions. The present study revealed that the negative regulation of the immune system was relatively inactivated in Hunner lesions compared to non-Hunner lesions. It is speculated that, when the immune response is activated in the early stages of inflammation, it becomes increasingly difficult to control, leading to the development of severe inflammation and ultimately the formation of Hunner lesions.

Clinical associations have been reported between IC and inflammatory bowel disease, as well as several autoimmune diseases [25]. Moreover, the development of various macrophage-targeting drugs for autoimmune disease therapy is actively being researched. However, no research exists on these targets in HIC that show similar immunologic features. In this study, in Hunner lesions, genes related to macrophage differentiation and polarization, such as *VSIG4*, *CD68*, *MAFB*, and *LIRB4*, were downregulated. Macrophages exist as phenotypically diverse subtypes, including classically activated macrophages (M1) and alternatively activated macrophages (M2). Macrophages change their functional phenotypes depending on diverse stimuli, including cytokines and growth factors, and macrophages exert both anti-inflammatory and pro-inflammatory influences through the secretion of these factors [26]. VSIG4 is expressed on the surface of a subset of macrophages and is involved in immune homeostasis by inhibiting T cell activation [27]. MAFB is a transcription factor that plays an important role in promoting anti-inflammatory macrophage polarization [28]. Moreover, we found that the cell fraction of M1 macrophages, which secrete pro-inflammatory cytokines, reactive nitrogen intermediates, and reactive oxygen species, was increased in Hunner lesions. In contrast, the cell fraction of M2 macrophages, which are involved in suppressing the immune response and promoting tissue repair, was decreased in Hunner lesions. In particular, IL13, a key cytokine that promotes M2 macrophage polarization, was downregulated in Hunner lesions, whereas IL11, a well-known regulator of fibrosis [16,17], was upregulated. Collectively, these results show that the regulation of the immune system plays a crucial role in the progression of HIC and suggest that macrophage polarization is a potential therapeutic target. In future studies, we plan to investigate the potential impact of macrophage dysfunction on the progression of HIC. While macrophage involvement has been observed, HIC’s classification as an autoimmune disease remains inconclusive. These investigations may yield insights into the development of targeted therapies for HIC.

The pathological features of HIC in the serosa and muscle layers remain largely unknown. Although a large body of previous work has revealed that fibrosis can induce muscle hyperplasia or hypertrophy in various diseases [29], studies on interstitial cystitis are limited because they have focused only on mucosal observation. In the present study, full-thickness bladder tissue was obtained from patients with HIC who underwent partial cystectomy, and the characteristics of the muscular layer/serous layer were analyzed. Our findings suggest that the progression of inflammation and fibrosis into the serosa/muscle layer can contribute to muscle thickness/hypertrophy. In one study, more than half of patients with IC had bladder wall thickening on CT scans [30], which is consistent with our results, suggesting that IC may be associated with bladder fibrosis. In our study, the mean number of urinary frequencies per day of the patients participating in the study greatly increased to approximately 27, and the maximum bladder capacity was considerably small, at 160 cc. Although it may be difficult to hold urine for a long time due to pain, the direct decrease in bladder volume due to the progression of fibrosis/muscle hypertrophy in end-stage patients may also have an effect.

Despite providing valuable insights into the pathophysiology of HIC, this study has several limitations. This study included only a small number of patients and healthy donors, which may not be representative of a larger population of individuals with HIC. This could limit the statistical power of the study and the generalizability of the results. Moreover, the study design was cross-sectional, which limits our understanding of the evolution of these molecular changes over time. Longitudinal studies are needed to better understand the progression and potential initiation of HIC. In addition, these findings were not validated in independent patient cohorts, which is important for confirming the reliability of the results. Our study did not aim to confirm the presence of immune cells through histology and staining. Therefore, we did not verify the histology of the immune cells as part of our research. Finally, the study identified several genetic pathways and alterations associated with HIC, but it did not analyze the functional impacts of these alterations.

In conclusion, Hunner lesions showed reduced anti-inflammatory and immunosuppressive properties compared to non-Hunner lesions, along with alterations in immune cell populations. This study suggests distinct molecular features between Hunner and nonHunner lesions. Furthermore, our observations suggest a potential link with macrophage polarization, as well as its relevance to the characteristics of autoimmune diseases. The findings of this study contribute to enhancing our understanding of the complex pathological processes in HIC and could provide crucial clues for the development of future therapeutic strategies for this disease.

## 4. Materials and Methods

### 4.1. Participants and Sample Collection

The eligible participants were patients aged >20 years with IC/BPS who had bladder pain with a visual analog scale (VAS) for pain of 4 or higher lasting for more than 6 months, as well as Hunner lesions confirmed through cystoscopy. Among the eligible patients, patients who underwent multiple endoscopic ablation treatments or patients with end-stage IC whose symptoms were so severe that endoscopic treatment was no longer possible and who decided to undergo supratrigonal cystectomy with augmentation cystoplasty were enrolled (*n* = 10). Hunner lesions and non-Hunner lesions (those that appeared grossly normal) were collected at full thickness from each patient’s bladder tissue and dissected into the mucosal layer and muscle/serosa layer to obtain a total of four samples per patient. In the control group (*n* = 4), bladder samples were obtained by dissecting normal bladder tissue into the mucosal and muscle/serosa layers of patients who underwent radical cystectomy or radical nephroureterectomy for bladder cancer or renal pelvic cancer. All patients with bladder cancer had a single mass, and tissues were obtained far from the tumor. In patients with renal pelvic cancer, we obtained a portion of the bladder tissue during bladder cuffing. All control tissue samples were reviewed by an experienced pathologist and were found to be normal. For the IC patient group, a 3-day voiding diary, O’Leary–Sant Interstitial Cystitis Questionnaire Symptom Index (ICSI), Interstitial Cystitis Questionnaire Problem Index (ICPI), Pelvic Pain and Urgency/Frequency Patient Symptom Scale (PUF), and visual analog scale for pain were collected. The study was approved by the Institutional Review Board (IRB No. 2018-03-025), and written informed consent was obtained from all patients prior to surgery.

### 4.2. Immunohistochemistry (IHC)

To detect M1- and M2-associated macrophage markers, IHC was performed on paraffin-embedded tissue samples. Briefly, the tissue samples were transferred onto slides, hydrated, and washed with phosphate buffer. The slides were then treated with serum or purified antibody solution specific for M1 or M2 macrophage markers, followed by a reaction with a developing agent until the target protein was visualized. After washing and dehydration, the tissue was mounted and examined under a microscope. Representative surface markers were used for antibody staining, targeting M1 macrophages CD80 (clone 37711, 1:50 dilution, Cat#MAB140, R&D Systems, Minneapolis, MN, USA) and MHC II (Clone CR3/43, 1:100 dilution, Cat#ab7856; Abcam, Cambridge, UK), and M2 macrophages CD163 (clone 10D6, 1:50 dilution, Cat#MA5-11458; Thermo Fisher Scientific, Waltham, MA, USA) and CD206 (polyclonal, 1:100 dilution, Cat#ab64693; Abcam). The number of positive cells was assessed in ten representative high-power fields (HPFs) for both Hunner and non-Hunner lesions. The average number of positive cells was recorded. In Hunner and non-Hunner lesions, the results were analyzed, and the number of positive cells was manually counted by a board-certified pathologist in ten representative HPFs and was recorded as the average (cell count/HPF).

### 4.3. Transcriptome Sequencing

RNA sequencing libraries were created following the instructions of the TruSeq RNA Exome Library Prep Kit (Illumina Inc., San Diego, CA, USA). Isolated total RNA was used in a reverse transcription reaction with random primers using SuperScript II reverse transcriptase (Invitrogen (Waltham, MA, USA), Thermo Fisher Scientific) according to the manufacturer’s protocols. RNA sequencing libraries were prepared via end-repair, 3′-end adenylation, adapter ligation, and amplification. These libraries were then sequenced using the 100-bp paired-end mode of the TruSeq Rapid PE Cluster Kit and TruSeq Rapid SBS Kit in Illumina HiSeq 2500 (Illumina).

### 4.4. Transcriptome Sequencing Analysis

The RNA sequencing reads were aligned to the human reference genome (GRCh38) using the 2-pass default mode of STAR (version 2.6.1) with the ENSEMBL annotation (version 98) and quantified as transcripts per million (TPM) using RSEM (version 1.3.1). To identify differentially expressed pathways, the package gene set variation analysis (GSVA) in R (version 4.0.3) software was used to calculate the pathway scores for hallmark gene sets in MSigDB. To identify differentially expressed genes (DEGs), low-expressed genes were filtered out, and differentially expressed genes were identified using DESeq2 package (version 1.40.2). To compare Hunner lesions and non-Hunner lesions from the same patients, a paired analysis was performed. Gene set enrichment analysis (GSEA) was performed with gene sets from the Gene Ontology Biological Process database. We quantified the fraction of the cell population using quanTIseq, a deconvolution method based on a RNA-seq derived signature, in quantiseqr R package (version 1.10.0).

### 4.5. Statistical Analysis

Descriptive statistics for continuous and categorical variables are presented as medians (interquartile ranges) and frequencies (%), respectively. Continuous variables between two groups were tested using the two-sample *t*-test or Wilcoxon rank-sum test according to normality. One-way analysis of variance (ANOVA) was used to compare the means of the three groups. Statistical significance was set at *p* < 0.05. All statistical analyses were performed using R version 4.0.3 (http://www.r-project.org, accessed on 10 October 2022).

## Figures and Tables

**Figure 1 ijms-25-00778-f001:**
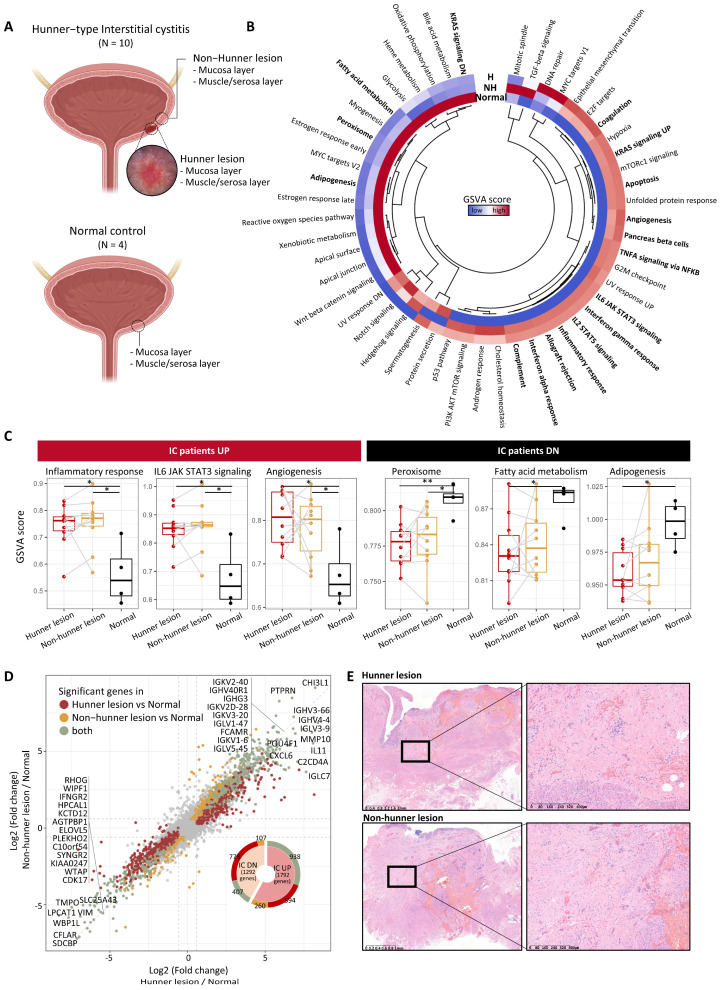
Fibrosis and immune activation in the mucosal layer of patients with Hunner-type interstitial cystitis. (**A**) Scheme of the overall study design. (**B**) Heatmap showing differences in pathway activities scored using GSVA among tissue types. The scores of each gene set were normalized using z-scores. Pathways that exhibited a statistical difference based on the one-way ANOVA test are indicated in bold. The relationship between the pathways is shown in the central dendrogram. (**C**) GSVA scores for some significant pathways are shown in (**B**). Statistical significance was estimated using the Wilcoxon rank-sum test and is marked with asterisks (*p* < 0.05: *, *p* < 0.01: **). Samples from the same patient are connected with a gray line. (**D**) Scatter plot showing fold change of genes in Hunner lesions compared to healthy tissues (*x*-axis) and non-Hunner lesions compared to healthy tissues (*y*-axis). Statistically significant genes are highlighted in three colors. Only the significant genes in the comparison between Hunner lesions and healthy tissues are colored dark red. Only the significant genes in the comparison between non-Hunner lesions and healthy tissues are colored in yellow. Genes that were significant in both comparisons are colored green, and the top 20 genes are indicated. (**E**) Representative full-thickness biopsies of Hunner and non-Hunner lesions showing prominent fibrosis (H&E staining). Abbreviation: H, Hunner lesion; NH, non-Hunner lesion; IC, interstitial cystitis.

**Figure 2 ijms-25-00778-f002:**
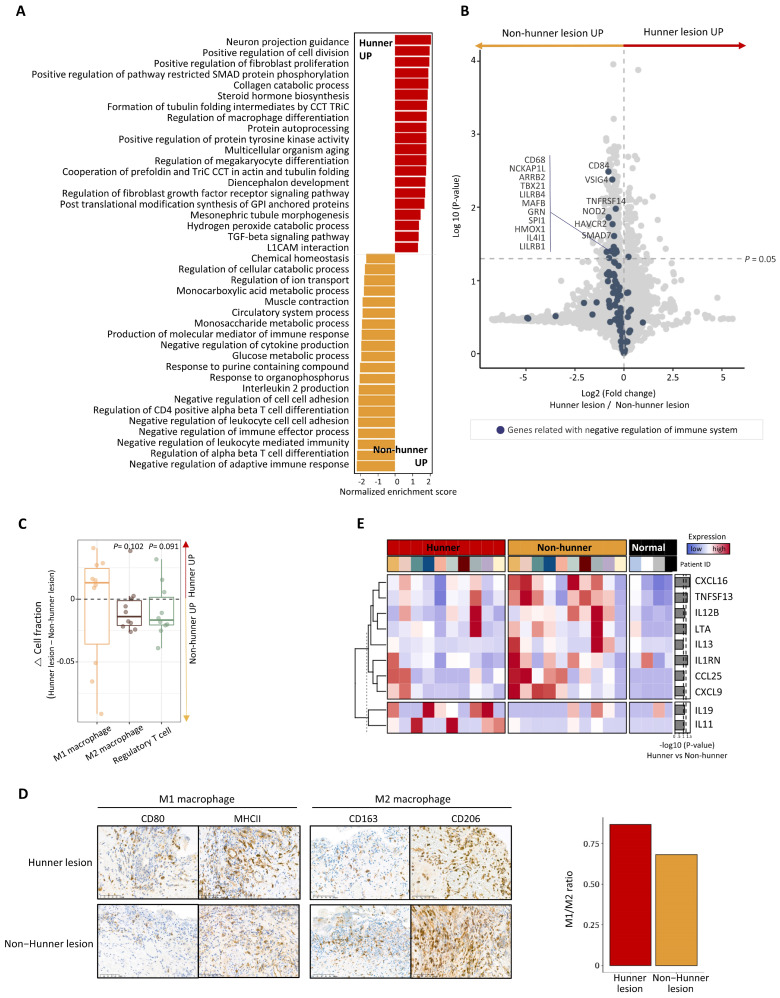
Differences between Hunner and non-Hunner lesions in the mucosal layer of the bladder in patients with Hunner-type interstitial cystitis. (**A**) Bar plot displaying the top enriched pathways in the comparison between Hunner and non-Hunner lesions. The normalized enrichment score (NES) was determined by gene set enrichment analysis (GSEA). The enriched pathways in Hunner lesions are shown in red and the enriched pathways in non-Hunner lesions are shown in yellow. (**B**) Volcano plot showing differentially expressed genes (DEG) between Hunner lesions and non-Hunner lesions, with blue dots representing genes related to negative regulation of the immune system. (**C**) Box plot showing the relative fraction of the cell population estimated by the RNA-seq deconvolution tool. The relative change in the cell fraction was calculated by subtracting the fraction of non-Hunner lesions from that of Hunner lesions. *p*-values were calculated using a one-way *t*-test. (**D**) (**Left**) Immunohistochemical staining showing elevated cell counts of M1 and M2 macrophages in Hunner lesions compared to non-Hunner lesions. (**Right**) Bar plot showing an increased ratio of M1 macrophages to M2 macrophages (M1/M2 ratio), defined as the ratio of the average number of cells stained with antibodies representing M1 macrophages to the average number of cells stained with antibodies representing M2 macrophages. (**E**) Heatmap showing the expression levels of the differentially expressed genes encoding cytokines and growth factors. Statistical significance was estimated using the Wilcoxon rank-sum test, and the *p*-values are illustrated on the right with bar plots.

**Figure 3 ijms-25-00778-f003:**
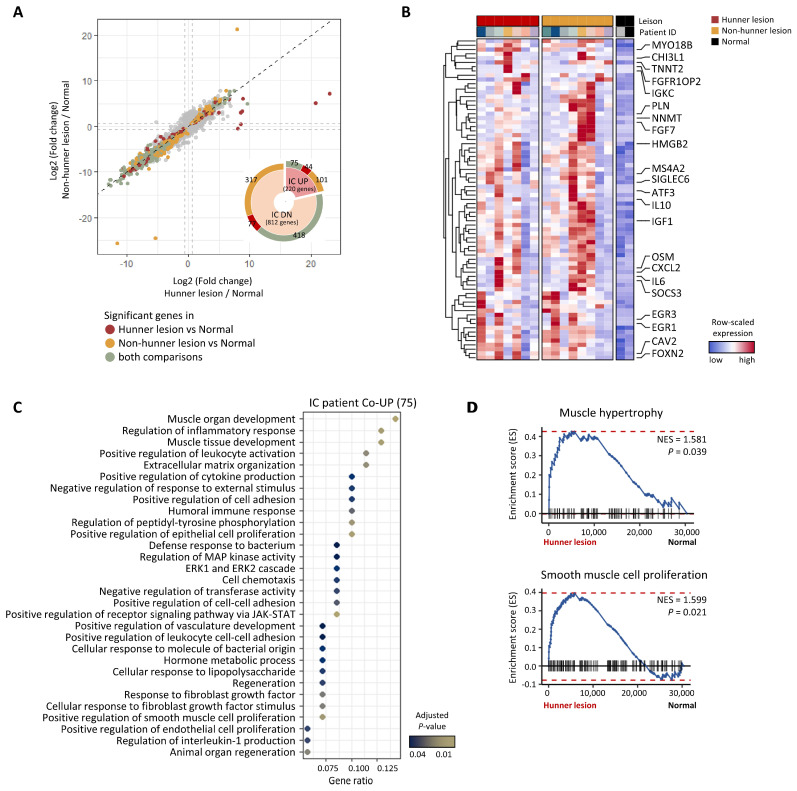
Thickening of serosal/muscular layer of patients with Hunner-type interstitial cystitis. (**A**) Scatter plot showing fold change of genes in Hunner lesions compared to healthy tissues (*x*-axis) and non-Hunner lesions compared to healthy tissues (*y*-axis). Statistically significant genes are highlighted in three colors. Only the significant genes in the comparison between Hunner lesions and healthy tissues are colored dark red. Only the significant genes in the comparison between non-Hunner lesions and healthy lesions are colored in yellow. The significant genes in both comparisons are shown in green. (**B**) Heatmap showing the expression levels of 75 genes that were significantly upregulated in both Hunner lesions and non-Hunner lesions compared to healthy tissues. (**C**) Dot plot representing the significantly enriched pathways of the 75 genes via over-representation analysis. (**D**) Gene set enrichment analysis (GSEA) plot of gene sets related to muscle hypertrophy/hyperplasia. *p*-values were calculated using pre-ranked GSEA with log-fold changes. No correction for multiple testing was performed, as only two gene sets were analyzed.

**Table 1 ijms-25-00778-t001:** Hunner-type interstitial cystitis patient characteristics (n = 10).

Characteristic	Value
Age (yr), median (IQR)	67.0 (55.5–72.6)
Sex, n (%)	
Male	2 (20.0)
Female	8 (80.0)
Number of endoscopic ablations, median (range)	3.0 (2–4)
Maximal cystometric capacity (ml), median (IQR)	160.0 (113.8–283.0)
Number of urinary frequency, mean (SD)	27.2 (11.1)
O’Leary–Sant Interstitial Cystitis Questionnaire (IC-Q), median (IQR)	
Symptom index	16.5 (14.8–18.3)
Problem index	15.0 (12.0–16.0)
Pelvic Pain and Urgency/Frequency (PUF) symptom scale, median (IQR)	
Symptom	17.5 (16.0–18.0)
Bothersome	9.0 (7.5–9.0)
Pain for VAS, median (IQR)	8.0 (6.0–9.0)

## Data Availability

The data from this study have been deposited in NCBI Sequence Read Archive (SRA) and are available under accession number PRJNA1002755.

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
