# Peer review of "Potential Role of Macrophage Polarization in the Progression of Hunner-Type Interstitial Cystitis"

_ijms, 2024, doi:10.3390/ijms25020778_

Round 1
Reviewer 1 Report
Comments and Suggestions for Authors
The authors present a novel observational study on full thickness bladder tissue of patients with Hunner type interstitial cystitis and healthy controls. Hunner lesions showed reduced expression 33 of anti-inflammatory and immunosuppressive factors compared to non-Hunner lesions, along with alterations in immune cell populations. This study is well designed; the results are interesting; the manuscript is well written with excellent figures. The findings enhance our scientific understanding of the pathophysiology of this condition and could potentially lead to therapeutic interventions. The manuscript might be improved by considering the following comments and questions.
Table 1. “Number of urinary frequency” is worded awkwardly. Does this mean number of voids per day? Does this include nocturia?
Results: 3.2: Paragraph 2: Sentence 2: The first part of the sentence states that the groups were similar, and the second half of the sentence shows comparisons w p< 0.05. This sentence should probably be divided in two with additional text explaining that there were significant differences.
Results: 3.4: Perhaps it is just my PDF, but it seems that the paragraph ends mid-sentence. The sentence that starts with “Additionally, we observed…” end with “however, this result was not.” This is line 238 on my PDF. Please ensure that this paragraph displays correctly on the final PDF.
Discussion: Please elaborate on the next research steps. Are there any specific drugs that could be proposed as potential options for treating HIC based on these findings? What additional studies would be needed to move these findings towards clinical translation?
Author Response
Response to Reviewer 1 comments
Table 1. “Number of urinary frequency” is worded awkwardly. Does this mean number of voids per day? Does this include nocturia?
Response: Thank you for your comments. Urinary frequency is the official terminology of the international continence society (ICS), and there appears to be no problem with its use. The total number of datetime and night-time voids is also included.
Results: 3.2: Paragraph 2: Sentence 2: The first part of the sentence states that the groups were similar, and the second half of the sentence shows comparisons w p< 0.05. This sentence should probably be divided in two with additional text explaining that there were significant differences.
Response: Thank you for your comments. One-way ANOVA was conducted to compare pathway scores among the three groups. The analysis revealed significant differences in several pathways related to the immune system among the groups. We visualized the degree of GSVA scores using colors and observed similar GSVA score patterns in pathways associated with the immune system between Hunner lesion and non-Hunner lesion.
I have revised the content of the manuscript as follows:
“Several pathways associated with the immune system, including interferon-gamma re-sponse (P = 0.002), inflammatory response (P = 0.005), allograft rejection (P = 0.006), inter-feron alpha response (P = 0.010), complement system (P=0.004), TNF-α signaling via NF-kB (P = 0.032), IL2/STAT5 signaling (P = 0.008), and IL6/JAK/STAT3 signaling (P = 0.003) exhibited significant variations in GSVA scores among Hunner lesion, Non-Hunner lesion, and normal groups. These pathways exhibited similar GSVA scores between Hun-ner lesion and non-Hunner lesion (Figure 1B, 1C).”
Results: 3.4: Perhaps it is just my PDF, but it seems that the paragraph ends mid-sentence. The sentence that starts with “Additionally, we observed…” end with “however, this result was not.” This is line 238 on my PDF. Please ensure that this paragraph displays correctly on the final PDF.
Response: Thank you for clarifying. The information was indicated at line 239 above the figure on the following page and has been reorganized for better readability.
Discussion: Please elaborate on the next research steps. Are there any specific drugs that could be proposed as potential options for treating HIC based on these findings? What additional studies would be needed to move these findings towards clinical translation?
Thank you very much for your kind comments. We appreciate your recognition of our work. Indeed, macrophages are recognized as key players in the pathogenesis of various autoimmune diseases. As outlined in our paper, we have also observed indications of the role of macrophages in Hunner type interstitial cystitis (HIC). However, it's important to acknowledge that HIC has not been definitively classified as an autoimmune disease to date. We agree with your assessment in this regard. Our research has provided some preliminary evidence of macrophage involvement in HIC, but further investigations are necessary to establish a clear link.
In our upcoming studies, we plan to delve deeper into the potential impact of macrophage dysfunction on the progression of Hunner type IC. We anticipate that the results of these studies will shed more light on the subject and, if positive, may lead to the identification of meaningful target drugs for this condition. Additionally, we are in the process of planning in vivo experiments related to the target gene of interest. While the specifics of these experiments have not been fully detailed in this paper, they represent an essential aspect of our ongoing research efforts to elucidate the mechanisms underlying our findings.
In discussion, line 314 : In future studies, we plan to investigate the potential impact of macrophage dysfunction on the progression of HIC. While macrophage involvement has been observed, HIC's clas-sification as an autoimmune disease remains inconclusive. These investigations may yield insights into the development of targeted therapies for HIC.

Reviewer 2 Report
Comments and Suggestions for Authors
The authors should be congratulated for their work. Interstitial cystitis is a rare and albeit unknown entity. It is mandatory to address more information on diagnostic, clinical, and therapeutic advances to better understand this bothersome disease. Despite the paucity of the sample, the current study specifically focused on the genetic differences between Hunner and non-Hunner lesions to improve the knowledge of the pathogenesis of these pathognomonic signs.
The manuscript is well-written and easily readable.
Minor comments to address:
- any data available on patients who experienced vulvodynia? Vulvodynia is a complex symptom pattern that is highly prevalent in young females (PMID= 19718938, 33901534, 37449291 ). Despite the presence of Hunner signs, how did the author discriminate these two often coupled conditions?
- Is any data available on other comorbidities of patients? Such as overactive bladder, OSAS, or cardiac impairment? All these diseases are related to urinary tract symptoms (PMID= 37167825, 33901534)
-Did you find any differences between the genetic expression of the core of Hunner lesions from the perimeter area?
Author Response
Response to Reviewer 2 comments
Thank you for your thoughtful comments regarding our recent research. We appreciate your engagement with our work and understand that all your inquiries are of significant importance. However, we would like to acknowledge that in the course of our study, some aspects you inquired about were challenging for us to address comprehensively.
We understand the value of addressing every question thoroughly and apologize for any limitations in our ability to do so in this study. We have taken note of your questions and will make efforts to address them more effectively in future research.
Any data available on patients who experienced vulvodynia? Vulvodynia is a complex symptom pattern that is highly prevalent in young females (PMID= 19718938, 33901534, 37449291). Despite the presence of Hunner signs, how did the author discriminate these two often coupled conditions?
Response: Unfortunately, we could not confirm whether vulvodynia was present, but our patients were clearly Hunner type IC patients who had undergone a median of 3 endoscopic treatments. These patients had frequent recurrences despite repeated procedures and their symptoms were so severe that endoscopic treatment was no longer possible. Also, please take into account that most of the patients were elderly, with a median age of 67 years.
Is any data available on other comorbidities of patients? Such as overactive bladder, OSAS, or cardiac impairment? All these diseases are related to urinary tract symptoms (PMID= 37167825, 33901534)
Response: The underlying diseases you mentioned may affect lower urinary tract symptoms, but this has not been investigated. The number of patients included in this study was so small, other comorbidities were expected to be difficult to derive meaningful results. As answered in the question above, these were patients with definite end-stage Hunter-type IC.
Did you find any differences between the genetic expression of the core of Hunner lesions from the perimeter area?
Response: In our recent study, we did not specifically analyze the core and perimeter areas of Hunner lesions. Our research did not include the analysis of these specific regions as it was not within the scope or objectives of our study. We focused on different research goals. We anticipate that future studies may provide more detailed information regarding the genetic characteristics of the core and perimeter areas of Hunner lesions.